# Fast and Selective Degradation of Biomass for Xylose, Glucose and Lignin under Mild Conditions

**DOI:** 10.3390/molecules28083306

**Published:** 2023-04-07

**Authors:** Shangzhong Zhang, Yi Duan, Changchang Teng, Hongdong Quan, Xiuguo Yang, Hongyan Li, Xiaohe Li, Lifeng Yan

**Affiliations:** 1Department of Chemical Physics, University of Science and Technology of China, Jinzai Road 96, Hefei 230026, China; 2Key Laboratory of Anhui for Tobacco Chemistry, Hefei 230088, China; 3Inner Mongolia Key Laboratory of Polyol Chemical New Material Enterprise, Chifeng Ruiyang Chemical Co., Ltd., Pingzhuang, Chifeng 024076, China

**Keywords:** lignocellulose, corncob, glucose, xylose, lignin

## Abstract

The conversion of lignocellulose into valuable chemicals has been recognized as the key technology in green chemistry. However, selective degradation of hemicellulose and cellulose with the production of lignin is still a challenge. Therefore, a two-step process has been developed to degrade corncob into xylose and glucose under mild conditions. At first, the corncob was treated with the lower concentration of zinc chloride aqueous solution (30–55 w%) at 95 °C with a short reaction time (8–12 min) and 30.4 w% (selectivity = 89%) of xylose obtained with a solid residue of the composite of cellulose and lignin. Next, the solid residue was treated with a high concentration of zinc chloride aqueous solution (65–85 w%) at 95 °C for about 10 min, and 29.4 w% (selectivity = 92%) of glucose can be obtained. Combining the two steps, the total yield of xylose is 97%, while glucose is 95%. In addition, high pure lignin can be obtained simultaneously, which was confirmed using HSQC studies. Furthermore, for the solid residue of the first-step reaction, a ternary deep eutectic solvent (DES) (choline chloride/oxalic acid/1,4-butanediol, ChCl/OA/BD) has been used to separate the cellulose and lignin efficiently, and high-quality cellulose (Re-C) and lignin (Re-L) were obtained. Furthermore, it provides a simple method to disassemble the lignocellulose for monosaccharides, lignin, and cellulose.

## 1. Introduction

The unsustainability of coal, oil, and natural gas and their negative ecological and environmental impacts have forced many researchers to develop new processes based on renewable materials [1]. Lignocellulose biomass is the most common renewable carbon resource and has received much attention for its conversion into biofuels and other platform chemicals.

Cellulose, hemicellulose, and lignin are the three major components of lignocellulosic biomass. However, it is difficult to separate them by simple chemical or biological treatments due to their special complex structure [2,3]. In previous studies, compared with cellulose, hemicellulose has fewer hydrogen bonds and exposes more active centers for degradation [4,5,6,7]. Hemicellulose can be converted into xylose, xylitol, furfural, or other chemicals [8,9,10]. Cellulose is difficult to dissolve and process due to its crystal structure and complex hydrogen bonds [11,12,13]. However, cellulose, as a macromolecular polysaccharide, can still be degraded into oligosaccharides under certain conditions and continue to degrade glucose and 5-hydroxymethylfurfural [14,15,16,17,18,19,20]. Lignin is a complex class of aromatic phenolic polymers. Numerous literature reports on lignin extraction with deep eutectic solvent (DES) or organic solvents [21,22,23].

In the treatment of lignocellulose, inorganic saline solutions are commonly used [24]. Compared with organic solvents and ionic liquids, the inorganic saltwater solution has stable and low vapor pressure properties, which are safer, environmentally friendly, and clean [25,26,27,28]. Corncob is a lignocellulosic biomass of interest. It is often used as a raw material for biomass conversion because of its high hemicellulose, cellulose content, low price, and ease of obtaining [29,30,31,32].

Alkali, acid, hot water pretreatment, and enzymatic hydrolysis are common methods for treating and degrading biomass, but these methods have advantages and disadvantages. Alkaline systems can partially degrade carbohydrates and produce some phenolic derivatives [33]. Acidic treatment may cause an increase in by-products, such as furfural, furroic acid, and 5-HMF [34,35,36]. The enzymatic solution method is relatively mild and environmentally friendly but may be high-cost and limited in production [37,38]. Hot water pretreatment is economical and environmentally friendly but has a longer reaction time and low yield [39,40].

In this work, we performed the selective isolation of xylose and glucose in a very simple two-step method by adjusting the concentration of ZnCl_2_ in the solution, using untreated corncob as the feedstock (Figure 1). Firstly, corncob powder was treated with a lower concentration of ZnCl_2_ to transfer the hemicellulose to xylose. Next, we separated the two solid-liquid phases to obtain a xylose-rich solution and a solid residue of cellulose and lignin. In the second step, a high-concentration of ZnCl_2_ treatment transforms the cellulose from the remaining filter residue into glucose, resulting in a glucose-rich solution and residual solid of lignin. By adding ammonia aqueous solution or passing through ammonia gas to the filtrate, the Zn^2+^ ion can form insoluble complex precipitation after it is filtered and dried, converted into Zn(OH)_2_, and the zinc ion is recycled. Zn^2+^ ions can be recovered to 97.94% throughout the process. It provides a simple and efficient method of treating biomass to produce xylose, glucose, and lignin. Furthermore, for the solid residue of the first-step reaction, a ternary deep eutectic solvent (DES) (choline chloride/oxalic acid/1,4-butanediol, ChCl/OA/BD) has been used to separate the cellulose and lignin efficiently, and high-quality cellulose (Re-C) and lignin (Re-L) were obtained. It provides a simple method to disassemble the lignocellulose for monosaccharides, lignin, and cellulose. Compared with similar studies, simple and environment-friendly reagents were used in this work. The reaction conditions were mild (lower temperature, shorter reaction time), and the yields of xylose and glucose were higher [41,42,43].

## 2. Result and Discussion

### 2.1. Effect of ZnCl_2_ Content

Here, corncob powder was treated directly with different concentrations of ZnCl_2_ at 95 °C for 10 min, and the yields of xylose and glucose were listed in Table 1. Interestingly, the xylose and glucose yield strongly correlates with the concentration of ZnCl_2_, as shown in Figure 1. At the lower concentration of ZnCl_2_ (50 w% of solution), the main product is xylose, while the yield of glucose is relatively low. However, as the concentration of ZnCl_2_ increases, the yield of xylose decreases while the yield of glucose gradually increases. This may be due to the concentration of ZnCl_2_ related to the coordination of zinc ions with oxygen atoms on the hydroxyl group of the hemicellulose and cellulose [44,45], and the lower concentration of treatment can only degrade hemicellulose because there are fewer hydrogen bonds.

Interestingly, due to many hydrogen bonds, cellulose has a dense crystalline region and was also degraded when treated at high concentrations (75 w% of solution). But as the treatment concentration continues to rise, the decrease of the yields of xylose and glucose may be due to the increased viscosity of the system, resulting in reduced mass transfer efficiency. Figure 2 shows the possible method for the formation of xylose and glucose by varying the concentration of ZnCl_2_. In the first step, the lower concentration of ZnCl_2_ converts hemicellulose into xylose (filtrate L1), while cellulose was better retained in R1. Next, at a high concentration of zinc chloride, the cellulose in R1 was degraded into glucose (L2), leaving the remaining lignin in the filter residue R2. Then, xylose in the filtation L1 was isolated as powder, while the solid R1 was furthermore treated by a DES for cellulose (Re-C) and lignin (Re-L) (Figure 3). 

### 2.2. Effect of Temperature

The reaction temperature is crucial in this work, related to whether cellulose and hemicellulose can be hydrolysed. As shown in Table 2 and Table 3, and Figure 4a,b, at room temperature, cellulose is difficult to hydrolysis even with a high concentration of acid and a long reaction time. But hemicellulose hydrolysis does not require a relatively high temperature and ZnCl_2_ concentrations. At 65 °C, treatment of the hemicellulose with the lower concentration yielded a considerable xylose yield (18.3 w%). However, with increasing temperature, the xylose production reaches the maximum of 30.4 w% at 95 °C. This may be because the side reaction is not intense at low temperatures. When the temperature is greater than 95 °C, then as the temperature continues to rise, we can observe that the product of the flask changes from light brown to black, indicating the degradation of monosaccharides or the formation of carbon dots [46,47].

Meanwhile, an increase in furfural as a byproduct was detected by HPLC. This indicates that the excessive temperature will accelerate the dehydration of the sugar, leading to the carbonization of the biomass and producing more humins. Interestingly, the low temperature did not convert the cellulose efficiently at high concentrations (65 °C, 6.9%), but the yield of glucose also peaked at 95 °C (29.4 w%). This may be due to the high concentration treatment having destroyed the cellulose structure, after which glucose production no longer requires high temperature. So in both low and high concentrations, we selected 95 °C as the reaction temperature.

It must be noted that due to the short reaction time, we need to preheat the reactants and the flask early. Otherwise, if following the conventional reaction method, the reactant has not been heated to the temperature of the oil bath when the reaction time ends. This situation will cause the reaction not to proceed effectively, and the yield is much reduced. Therefore, we mixed and stirred a certain mass of ZnCl_2_ and corncob powder in a flask and next placed the flask in a 95 °C oil bath heater. We next measured the temperature of the in-bottle reactant with a thermometer. We quickly added hydrochloric acid and deionized water when 80 °C was reached and started the timing. At this point, the effects of dissolved heat and agitation will rapidly increase the temperature inside the flask to 93–95 °C and reach thermal equilibrium. Thus, the truth and accuracy of the reaction temperature are guaranteed.

### 2.3. Effect of Time

Reaction time is another key parameter; the results are shown in Table 4 and Table 5. As shown in Figure 4c,d, when treated with lower concentrations of ZnCl_2_, glucose yield was consistently low. It increased slowly over time due to the limited effect of Lewis acid on cellulose. However, the lower concentration treatment affects hemicellulose much more strongly and can be significantly converted into xylose in a short time (4 min, 17.5 w%). Therefore, the xylose yield was maximized at the reaction time of 10 min. At high concentrations of treatment, xylose yield was at low levels because hemicellulose had been removed, while glucose production also decreased after reaching a maximum. It may be because the long reaction time aggravates sugar dehydration and carbonization. To achieve a satisfactory separation effect and a maximum sugar yield, the low and high concentrations were treated with 10 min as the reaction time. Combining the two steps, the total yield of xylose is 97%, while glucose is 95%.

### 2.4. Characterization of the Intermediates

The shape and microstructure change of the corncob during the two-step treatment were measured by means of SEM. As shown in Figure 5, in the images of the Corncob, we can see the packing of the irregular shapes, which is due to the hemicellulose, cellulose, and lignin being entangled with each other. However, after ZnCl_2_ treatments, in the SEM images of R1 (Figure 5b–d), the apparent axial rod-shaped arrangement can be observed at all scales. This morphology is the characteristic crystal structure of cellulose. The cellulose structure in R1 is not well disrupted. However, in R2 (Figure 5e,f), we only observed an irregular porous structure, the remaining lignin, after removing the cellulose. This evidence confirmed that we achieved the purpose of isolating biomass and separate extraction of xylose and glucose, and lignin was also obtained simultaneously.

The FT-IR characterization was performed for the MCC, Corncob, R1, and R2, as shown in Figure 6a. At 3570 cm^−1^, R2 signals much stronger than R1 and corncob because of the characteristic absorption of the phenolic hydroxyl group of the benzene ring in the lignin structure. Because the continuous two-step reaction removed the main components of hemicellulose and cellulose, the amount of lignin content in the second filter residue R2 was increased, so the characteristics of lignin were more obvious. Similarly, at 1609 cm^−1^, it is also a significant signal for R2, the standard absorption peak for the vibration of the benzene ring skeleton in lignin. The peak at 1036 cm^−1^ is due to the stretching vibration of C-O in the polysaccharide. Since almost all of the polysaccharide in R2 has been transformed, the signal for this peak in R2 is somewhat weaker than that in R1 and Corncob. The peaks at 2918 cm^−1^ are generated by C-H extension vibrations in cellulose, shown in the MCC, Corncob, and R1 FTIR spectra. The cellulose has been removed in R2, so this signal is not obvious in the figure [48,49,50].

Figure 6b shows the XRD patterns of MCC, corncob, R1, and R2. They all had signals at the characteristic position of the cellulose (2θ = 22.4°), and the crystallinity (CrI) was 61.71%, 42.01%, and 54.13%, respectively, and the signal was insignificant in R2 [51,52]. This indicates that the lower concentration treatment can effectively remove the hemicellulose and retain the cellulose while the CrI of the cellulose also increases (from 42.01% to 54.13%). No obvious characteristic cellulose absorption peak in R2 occurs because the high concentration of treatment degraded the cellulose into glucose, as confirmed by HPLC.

### 2.5. Recovery of Zinc Ions

We used the coordination of Zn^2+^ and NH^4+^ to separate and recover the Zn^2+^ ion [53]. In this system, an ammonia aqueous solution or ammonia gas was added, and Zn^2+^ and NH^4+^ could combine to form a complex and an insoluble complex. S1 and S2 are obtained in a filtered manner, and S1 S2 is heated and dried and converted to S3. XRD studies reveal that the main component of S3 is Zn (OH)_2_ (Figure 7). S3, Zn (OH)_2_, and a mixture of both showed highly similar features on the XRD profile. Each absorption peak on the graph line of Zn (OH)_2_ can correspond to the same position in S3, which indicates the high purity of the main component Zn (OH) _2_ in S3. In addition, in the XRD patterns, the Zn (OH)_2_ and S3 graph lines only differ at 2θ = 8.1°, which may be due to the anisotropy.

The precipitation mode of feeding through ammonia gas and adding ammonia water had a similar effect, obtaining R(Zn^2+^) = 97.94% and R(Zn^2+^) = 96.71%, respectively, probably because the ammonia water introduced more solvents, and the quality of the precipitation was reduced. Notably, although precipitation occurs immediately after the addition of the reagent, placing the filtrate for a longer time is also necessary (in a 50 mL glass beaker for more than 24 h). According to the principle of precipitation dissolution equilibrium, precipitation of the filtrate several times can reduce the concentration of Zn^2+^ in the liquid and improve R(Zn^2+^).

In addition, the residual Zn^2+^ ion in the as-prepared xylose and glucose were also measured by means of ICP. They are 119.44 mg/L and 64.61 mg/L, respectively. The calculated purity of xylose was 98.96%, and that of glucose was 99.41%.

### 2.6. The Formation of Lignin-R2

The R1, obtained by the lower concentration treatment, mainly comprises cellulose and lignin. After R1 was treated by step 2, most of the cellulose was degraded, and the residue should be lignin. The NMR (HSQC) studies were carried out to analyze its purity. For the ^1^H^−13^C HSQC NMR spectroscopy using a Bruker AVIII 400 MHz spectrometer equipped with a DCH cryoprobe, around 50 mg of Re-L or R2 was dissolved in 1.2 mL of DMSO-d6. The spectral widths were 12 ppm (11 to 1 ppm) and 220 ppm (from 200 to 20 ppm) for the ^1^H and ^13^C dimensions, respectively. The peak of the solvent (DMSO-d6) was used as a reference point for the internal chemical shift. Data of spectra were processed using standard MestReNova-NMR2010 software.

Figure 8 shows the 1H-13C HSQC NMR spectrum of the as-obtained R2, and many hydroxyl corresponding signals are displayed. The signals are mainly aromatic region signals corresponding to guaiacyl (G), *p*-hydroxyphenyl (H), and syringyl (S) units of lignin. The side chains in the HSQC map are located in (δC/δH 50–95/2.8–4.5 ppm), and the aromatic regions are located in (δC/δH 100–135/6.2–7.8 ppm). The signal at δC/δH 55.65/3.83 ppm corresponds to the methoxy group and several β-O-4′aromatic ether bonds. In the spectrograph, the main position of the C_α_–H_α_ bonds of the A units connected to the G units and S units, respectively, is located in δC/δH 65.18/4.12 and 66.73/4.24 ppm. The C_γ_–H_γ_ correlations in structures A, B, and C can be distinguished at δC/δH 60.30/3.54, 60.59/3.41, 67.42/3.39 ppm, respectively. The signal of the C_β_–H_β_ bond between structures B and C overlaps with that of the methoxy group, which are located in δC/δH 54.67/3.44 and 53.63/3.10 ppm, respectively. The C_2,6_–H_2,6_ bonds of the S′ unit are mainly at δC/δH 111.14/7.11 ppm position, which is a relatively weak signal. The C_5_–H_5_ and C_6_–H_6_ bonds of the G structural unit were observed at δC/δH 116.22/6.90 and 116.12/6.79 ppm positions, respectively. Furthermore, the peaks of C_2,6_–H_2,6_ and C_3,5_–H_3,5_ bonds of the H structural unit were observed at δC/δH 115.38/6.63 and 130.80/7.53 ppm positions. The above observations indicate that the obtained R2 has characteristic structures and bonds of lignin, with main components of lignin.

### 2.7. The Formation of Re-C and Re-L

R1 can be treated by DES to yield cellulose Re-C and lignin Re-L, respectively. Next, we performed an XRD characterization (Figure 9) of Re-C, and Re-C has cellulose-characteristic absorption peaks at 2θ = 22.4°. In addition, the crystallinity of the Re-C (53.69%) was significantly higher than that of the corn cob (42.01%) and was similar to that of the filter residue R1 (54.13%). This indicates that the lower concentration treatment selectively degraded hemicellulose and well-preserved the structure of cellulose. Moreover, the method provides new ideas and insights into the selective separation of lignocellulose.

Figure 10 shows the 1H-13C HSQC NMR spectrum of the Re-L. It also shows the typical structural feature of lignin, similar to R2. (the method is the same as Section 2.7)

Interestingly, a clear distinction is shown between them, with a clear signal of C_2,6_–H_2,6_ bonds of the S unit located in δC/δH 103.93/6.69 ppm, appearing in Re-L, but not in R2. This distinction may be due to conditional differences between the two reactions. Moreover, the spectra of Re-L also showed structural information of lignin, demonstrating the lignin purity. The signal at δC/δH 56.18/3.84 ppm corresponds to the methoxy group and several β-O-4′aromatic ether bonds. In the spectrograph, the main position of the C_α_–H_α_ bonds of the A units connected to the G units and S units, respectively, is located in δC/δH 65.10/4.11 and 66.55/4.25 ppm. The C_γ_–H_γ_ correlations in structures A, B, and C can be distinguished at δC/δH 60.65/3.41, 62.18/3.39, 67.32/3.40 ppm, respectively. The signal of the C_β_–H_β_ bond between structures B and C overlaps with that of the methoxy group, which are located in δC/δH 53.62/3.11 and 54.67/3.44 ppm, respectively. The C_2,6_–H_2,6_ bonds of the S′ unit are mainly at δC/δH 111.12/7.11 ppm. The C_5_–H_5_ and C_6_–H_6_ bonds of the G structural unit were observed at δC/δH 116.12/6.78 and 115.16/6.78 ppm positions, respectively. Furthermore, the peaks of C_2,6_–H_2,6_ and C_3,5_–H_3,5_ bonds of the H structural unit were observed at δC/δH 115.39/6.52 and 130.79/7.46 ppm positions.

According to Figure 11, in the whole process, we obtained more than 60 w% of the monosaccharides (xylose and glucose), which is an exciting thing. At the same time, xylose and glucose get a good separation effect. Furthermore, in the final product, we obtained 15.3 w% of the lignin, enabling the efficient utilization of the lignocellulose. In addition, the solid residue of the first-step reaction can also be efficiently fractionated into high-quality cellulose and lignin by means of a ternary DES system.

## 3. Materials and Methods

### 3.1. Materials

Corncob from Inner Mongolia, China, were placed in sealed plastic bags to dry naturally for about three months, ground to powder, and screened through 80 mesh before being used. The weight loss of corncob powder was less than 0.01 g/g in three months. The main components of the corncob were determined by the NREL LAP method, resulting in 35% cellulose content, 34% hemicellulose content, and 19% lignin content.

Anhydrous zinc chloride, concentrated hydrochloric acid (12 mol/L), aqueous ammonia (25–28 w%), ethyl acetate, choline chloride, oxalate dihydrate, 1,4-butanediol, glucose, xylose, furfural, 5-HMF were all purchased from Sinopharm Chemical Reagents Co., LTD. (Shanghai, China). Microcrystalline cellulose, purchased from Shanghai Aladdin reagent company (Shanghai, China). Ammonia gas, analytical pure, purchased from Anhui Hefei Hengxing Industrial Gas Co., Ltd. (Hefei, China), stored in cylinders. The flow rate is 1 mL/s.

Chromatographic grade acetonitrile for HPLC was purchased from Merck (Rahway, NJ, USA). Ultrapure water was prepared via a Milli-Q system (18.2 MΩ, Millipore, Burlington, MA, USA). The hydrochloric acid (2 mol/L) used in the experiment was obtained by diluting the concentrated hydrochloric acid and ultrapure water.

### 3.2. Selective Degradation of Hemicellulose: Lower Concentration ZnCl_2_ Solution Treatment

10 g of anhydrous zinc chloride was put in a 50 mL flask, and then 1.5 g of crushed corncob powder was added under magnetical stirring to mix them evenly. Next, the mixture was heated by a water bath with a set temperature of 95 °C. The temperature of the reactant in the flask was measured using a thermometer. When the temperature of the reactant reached 80 °C, 5 mL of ultrapure water and 5 mL of hydrochloric acid (2 mol/L) were added under stirring and maintaining the reaction temperature at 90–95 °C using the dissolved heat of zinc chloride. The timing was started after the addition of water and hydrochloric acid was completed. After 10 min of the reaction, the flask was removed from the heater, and 30 mL of cold ultrapure water was added to stop the reaction. The products inside the flask were filtered to generate residue R1 and filtrate L1, respectively. The L1 components were determined by HPLC (Shimadzu, LC-20D). Byproduct species was determined by adding 20 mL of ethyl acetate to extract L1, resting for 2 h, and detecting the organic phase C1 by the HPLC method. Then 0.5 g activated carbon was added to the aqueous phase H1 for adsorption of pigment from corn cob and decolorization, under stirring at 65 °C for 0.5 h. Next, it was filtered to obtain the filtrate H2. Ammonia was injected into H2 to adjust pH = 8 to allow the zinc ions to precipitate. Three repeats were carried out to completely remove the zinc ions in H2. The filter residue S1 and the xylose-rich filtrate H3 were treated separately, and the H3 was distilled under reduced pressure to remove the water, and the resulting white solid was the xylose.

### 3.3. Selective Degradation of Cellulose: High Concentration ZnCl_2_ Solution Treatment

15 g of anhydrous zinc chloride was added in a 50 mL flask, and then add the crushed and dried residue R1, and mixed under magnetic stirring at room temperature. The mixture was heated by a water bath with a set temperature of 95 °C. The temperature of the reactant in the flask was measured using a thermometer. When the temperature of the reactant reached 80 °C, 5 mL of hydrochloric acid (2 mol/L) was added with a pipette gun under stirring, and maintain the reaction temperature at 90–95 °C using the dissolved heat of zinc chloride. The timing was started after the addition of hydrochloric acid was completed, and after 10 min of the reaction, the flask was removed from the heater, and 30 mL of cold ultrapure water was added to stop the reaction. The products inside the flask were filtered to generate residue R2 and filtrate L2, respectively. The L2 components were determined by HPLC. The 0.5 g activated carbon was added to the aqueous phase L2 for adsorption of pigment from corn cob and decolorization, heated, and stirred at 65 °C for 0.5 h, and it was filtered to obtain the filtrate H4. Ammonia was injected into H4 to adjust pH = 8 to allow zinc ions to precipitate and filter. The process was repeated three times to completely remove the zinc ions in H4. The filter residue S2 and the glucose-rich filtrate H5 were treated separately, and the H5 was distilled under reduced pressure to remove the water, and the resulting white solid was the glucose.

### 3.4. Fractionation of Lignocellulose in DES

The solid residue of the first-step reaction is a complex of cellulose and lignin, which can be fractionated using the green solvent of DES [54]. Here, a ternary DES of oxalic acid dihydrate (OA, 0.1 mol), choline chloride(ChCl, 0.1 mol), and 1,4-butanediol (BD, 0.2 mol) were prepared and was used as the solvent to separate the solid residue in an oil bath at 110 °C for 5 h under continuous stirring. After the reaction, the mixture was added to 100 mL of anti-solvent (ethanol:water = 3:7) under stirring for 2 h. The filter residues were washed thrice with 100 mL anti-solvent and dried to obtain cellulose Re-C. Then, ethanol in the filtrate was removed by rotary evaporation, and the filtrate was precipitated in 100 mL of dilute hydrochloric acid (pH = 2) to obtain lignin Re-L.

### 3.5. Method of Recovery of Zinc Ion

The residues S1 and S2 were dried in the vacuum drying chamber for 24 h at a temperature of 80 °C. The dried white solid S3 is insoluble in water, which was measured by XRD. Its XRD profile is the same as that of Zn(OH)_2_, indicating that the main component of S3 is Zn(OH)_2_. Then, S3 was weighed to calculate the zinc ion recovery.

### 3.6. Residual Zinc Ions in the Products

The residual zinc ion in xylose and glucose were detected using an Inductively coupled plasma spectrometer (ICP), model number PerkinElmer Optima 7300 DV. The obtained xylose and glucose were dissolved in ultrapure water and fixed to 50 mL.

### 3.7. Calculation Method of the Parameters

The selectivity of xylose: S_x_ = Y_mxi_/(Y_mxi_ + Y_mgi_), and the selectivity of glucose: S_x_ = Y_mgi_/(Y_mxi_ + Y_mgi_), where I = 1 or 2, indicates the first step and second step, m is the relative mass, while m_1_, m_2_, and m_3_ are the mass of corncob, hemicellulose, and cellulose. The crystallinity (crystallinity index, CrI) of corncob, MCC, R1, and R2:$$\mathrm{CrI}=\frac{I_{002}-I_{am}}{I_{002}}$$
where $I_{002}$ is the intensity of the diffraction peak at 2θ = 22.4°, and $I_{am}$ represents the amorphous portion at 2θ = 18°.

### 3.8. SEM Characterization of Corncob and R1, R2

The microstructure of products was observed by scanning electron microscope (SEM, Gemini SEM 500). Before testing, products were replaced with water and then freeze-dried.

### 3.9. FT-IR Characterization of Corncob, MCC, R1, and R2

Fourier transforms infrared (FTIR) spectra were recorded on a Bruker vector-2 spectrophotometer in the 400–4000 cm^−1^.

### 3.10. XRD Characterization of Corncob, MCC, R1, and R2

XRD analysis was performed on a Philips X’ Pert PRO SUPER X-ray diffractometer with Cu Ka radiation (l = 1.54056 Å).

### 3.11. Determination of Xylose and Glucose in the Filtrate L1 and L2 by Means of HPLC

Monosaccharides in the filtrates were quantitatively analyzed by means of high-performance liquid chromatography (Shimadzu) equipped with a refractive index detector (RID-10A) and NH_2_ column (4.6 × 250 mm^2^, purchased from Agilent,880952-708) at 40 °C. The mixed solution of acetonitrile and deionized water (70:30 *v*/*v*) was used as the mobile phase at a flow rate of 1.0 mL min^−1^.

### 3.12. Determination of Furfural and 5-HMF in the C1 by Means of HPLC

The dehydration products (5-HMF and furfural) of the monosaccharides were determined using HPLC(Shimadzu) with a C18 column (4.6 × 250 mm^2^, purchased from Agilent) at 40 °C and a UV–vis detector (SPD-20A) at 210 nm. The mixed methanol and ultrapure water (30:70, *v*/*v*) were used as the mobile phase at a flow rate of 0.7 mL min^−1^. Due to the low content, only furfural and 5-HMF were qualitatively examined in this work.

### 3.13. The XRD Characterization of Recovered Zn(OH)_2_

XRD analysis was performed on a Philips X’Pert PRO SUPER X-ray diffractometer with Cu Ka radiation (l = 1.54056 Å). Using the XRD method to detect the recovered Zn(OH)_2_ and the standard Zn(OH)_2_.

## 4. Conclusions

Degradation of lignocellulosic biomass is a key step to disassembling the complex structure of lignocellulose. Here, the lower concentration of ZnCl_2_ aqueous solution (30–55 wt %) was an efficient system to degrade hemicellulose to xylose (97%) selectively. At the same time, cellulose and lignin have less change. The selective degradation of hemicellulose makes the lignocellulose easier to degrade or fraction. For degradation, a high glucose yield (95%) can be achieved using a high concentration of ZnCl_2_ aqueous solution. In contrast, high-quality cellulose can be isolated when a ternary DES (ChCl/OA/BD) system is used. At the same time, higher purity lignin can be obtained by both treatment methods. It provides a new idea and route for the efficient separation and utilization of lignocellulosic biomass.

## Data Availability

Data are available from the authors if required.

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
