# Peer review of "Fast and Selective Degradation of Biomass for Xylose, Glucose and Lignin under Mild Conditions"

_molecules, 2023, doi:10.3390/molecules28083306_

Round 1
Reviewer 1 Report
Utilizing biomass as a valuable chemical is an important agenda of this era. Zhang et al introduced the process of obtaining glucose and xylose by treating corncob with zinc chloride under various conditions. This study is judged to be an overall solid study ranging from the analysis of the product obtained from corncob to the characteristics of the processing process using SEM and sample analysis using X-rays. I think it's something that can be interesting to see in the related field. I support the publication of the paper after some revisions below.
1. Investigate studies similar to the author's findings, and have experimental and resultant comparative discussions with them. A relative comparison would make the findings of this study more scientifically sound.
2. The Methods section does not contain enough information to reproduce the study. In particular, for sections 3.5, 3.6, and 3.8-3.13, information on sample preparation should be added.
3. line 98-101 "This may be due to the concentration of ZnCl2 related to the coordination of zinc ions with oxygen atoms on the hydroxyl group of the hemicellulose and cellulose, and the low concentration of treatment can only degrade the structured more scattered hemicellulose ." If there are previous references to support this, they should be added.
4. There are many typos in spaces or units in the manuscript. Authors should read and revise carefully.
Minor,
1. line 396: remove 'In conclusion'
2. There is a reference with missing page information.
3. Figure 2: Authors need to correct arrows overlapping figures or text.
4. Figure 3: 'decolorization ,remove zinc ions' should be 'decolorization, remove zinc ions'It's not essential, but for figure 3 it would be nice to improve the photo quality.
5. Figure 4: 'Temperature(°C)' should be 'Temperature (°C)' and 'Time(min)' should be 'Time (min)'
Author Response
Comments and Suggestions for Authors
Utilizing biomass as a valuable chemical is an important agenda of this era. Zhang et al introduced the process of obtaining glucose and xylose by treating corncob with zinc chloride under various conditions. This study is judged to be an overall solid study ranging from the analysis of the product obtained from corncob to the characteristics of the processing process using SEM and sample analysis using X-rays. I think it's something that can be interesting to see in the related field. I support the publication of the paper after some revisions below.
Reply: Thanks, we are appreciated to your positive comments.
- Investigate studies similar to the author's findings, and have experimental and resultant comparative discussions with them. A relative comparison would make the findings of this study more scientifically sound.
Reply: Thank you for your suggestion. We have added a discussion of similar work and related references to the original article, at the end of the introduction.
- The Methods section does not contain enough information to reproduce the study. In particular, for sections 3.5, 3.6, and 3.8-3.13, information on sample preparation should be added.
Reply: We have updated information on equipment, characterization methods, and samples at relevant locations.
- line 98-101 "This may be due to the concentration of ZnCl2 related to the coordination of zinc ions with oxygen atoms on the hydroxyl group of the hemicellulose and cellulose, and the low concentration of treatment can only degrade the structured more scattered hemicellulose ." If there are previous references to support this, they should be added.
Reply: We have added Ref. 45-46 at the corresponding location to support this.
- There are many typos in spaces or units in the manuscript. Authors should read and revise carefully.
Reply: Thank you for your suggestion. We have reread it carefully and revised it item by item according to your suggestion.
Minor,
- line 396: remove 'In conclusion'
Reply: It was removed
2. There is a reference with missing page information.
Reply: The missing page information were added.
3. Figure 2: Authors need to correct arrows overlapping figures or text.
Reply: It was improved
4. Figure 3: 'decolorization ,remove zinc ions' should be 'decolorization, remove zinc ions'It's not essential, but for figure 3 it would be nice to improve the photo quality.
Reply: They were improved.
5. Figure 4: 'Temperature(°C)' should be 'Temperature (°C)' and 'Time(min)' should be 'Time (min)'
Reply: They were corrected.
Reviewer 2 Report
Interesting direction of research. As noted in my review document, there is considerable lack of clarity on several points. The methods used need to be clarified. Moisture needs to be accounted for in the biomass and products. How biomass was stored to keep it dry or moisture constant is not stated. I recommend clarifying what figures 2 and 3 are trying to say, as it is not clear. Having them next to each other makes it much less clear that figure 3 is specifically just for the DES process. Finally, I strongly recommend giving some thought to seeming to claim industrial value, without a clear analysis of process chemical recovery.

Author Response
Comments and Suggestions for Authors
Interesting direction of research. As noted in my review document, there is considerable lack of clarity on several points. The methods used need to be clarified. Moisture needs to be accounted for in the biomass and products. How biomass was stored to keep it dry or moisture constant is not stated. I recommend clarifying what figures 2 and 3 are trying to say, as it is not clear. Having them next to each other makes it much less clear that figure 3 is specifically just for the DES process. Finally, I strongly recommend giving some thought to seeming to claim industrial value, without a clear analysis of process chemical recovery.
Reply: Thanks, we are appreciated to your positive comments. Thank you very much for your careful reading and advice. I think it is very helpful for the work. We have corrected the grammatical and expressive mistakes you mentioned. Figures 2 and 3 are improved to make them clear.
(1)110:Why are the filtrates being processed over activated carbon? What purpose does this have, and what is being absorbed?
Reply: The activated carbon is used to de-color the corn cob in the filtrate. Before the adsorption, the filtrate is dark yellow, and then the filtrate is colorless
(2)110: how are you recovering xylose and glucose by “distillation”
Reply: Water was removed by reduced-pressure distillation, and xylose and glucose were obtained as residual solids
(3)112: What this figure is referring to is unclear until much later in the paper. In this location it is extremely confusing.
Reply: Thank you for your suggestion. These symbols represent the products obtained in each process, and the article uses them to distinguish between them
(4)126:“high concentration of acid” was additional HCl added? The amount of hydrochloric acid added has been described in Table2, Table3, Table4 and Table 5
Reply: Thank you for your suggestion. The amount of additional hydrochloric acid is indicated in title of Table2, Table3, Table4 and Table 5
(5)127: The concentrations of ZnCl2 used throughout are quite high. I would not consider any of them “low”
Reply: Thank you for your suggestion. The concentration in the first step is lower than in the second step, so it is more appropriate to use the comparison level. "low" has been replaced with "lower" in the article
(6)129: (again at 130,139, 164, 170, 285, 288): weight % of what (biomass? Xylan? Theoretical xylose?)
Reply: The denominator for all quality fractions is total corncob mass which has been restated in the relevant chart heading.
(7)135: “carbon dots”? what is meant by this, and is there data to support this?
Reply: Thank you for your suggestion. Carbon dots (CDs), as a new type of fluorescent carbon-based nanomaterial, have attracted considerable attention since first reported in 2004. The work of our team and others has demonstrated that the generation of carbon dots is related to the carbonization of biomass, and the relevant references have been inserted in the original text.[1,2]
(8)143: How do you know no reaction occurred during the heat-up with ZnCl2 and biomass?
Reply: Because during the preheating of ZnCl2 and biomass, they are all solid, not in the same phase, and no liquid participates.
(9)143-148: what kind of water / oil bath and reaction vessel were used? 115 deg. C suggests a pressure flask and oil bath?
Reply: A normal pressure oil bath heating pot and a 50ml glass round bottom flask were used, with no obvious boiling and vaporization at 115 degrees Celsius, this may be because the boiling point of aqueous zinc chloride solution is higher than water which we thought it was feasible in terms of safety.
(10)186: MCC? This is never explained / sourced? Is this microcrystalline cellulose, and if so, why does it have a lignin peak in the FTIR?
Reply: MCC is a microcrystalline cellulose purchased from Shanghai Aladdin reagent company, the source of which has been added to this article. You mentioned that it has a lignin absorption peak in FTIR. This may be due to the instrument operating conditions or the lack of thorough purification by the reagent company, but we do not think this affects the identification of its main features.
(11)211 (and 337): was aqueous or gaseous ammonia used? If aqueous, at what concentration
Reply: Thank you for your suggestion. The concentration of aqueous ammonia used is 25-28 w%, purchased from Sinopharm Chemical Reagents Co., LTD. (Shanghai, China), the source of which has been added.
(12)223: “placing the filtrate” is unclear
Reply: The filtrate was left standing in a 50 ml glass beaker for more than 24 h, and this information has been added to the original text.
(13)223-225: I am unclear what is meant by this sentence
Reply: A longer standing time allows more zinc ions to bind to ammonium ions and precipitate as a solid. Similarly, the concentration of zinc ions in the first filtrate is still high due to the precipitation-dissolution equilibrium. We add ammonia water or ammonia gas to each of the filtrates and repeat this many times to recover more zinc ions.
(14)226-228: These values of purity are inconsistent with the earlier yield numbers (Table 4&5)
Reply: Thank you for your suggestion. These purity values refer to the purity in the final obtained xylose and glucose crude product (obtained by subtracting the zinc ion mass in the last filtrate from the total mass of xylose or glucose), whereas Table 4 & 5, the yield refers to the value of xylose and glucose obtained with the quality of corncob as the denominator.
(15)234: No information is given on the method and equipment used for this analysis. NMR maker and MHz rating, probe, solvent system, preparation should be included or referenced.
Reply: Thank you for your suggestion. “For the 1H−13C HSQC NMR spectroscopy using a Bruker AVIII 400 MHz spectrometer equipped with a DCH cryoprobe, around 50 mg of Re-L or R2 was dissolved in 1.2 mL of DMSO-d6. The spectral widths were 12 ppm (from 11 to 1 ppm) and 220 ppm (from 200 to 20 ppm) for the 1H and 13C dimensions, respectively. The peak of the solvent (DMSO-d6) was used as a reference point for the internal chemical shift. Data of spectra were processed using standard MestReNova-NMR software.” This information has been added to the text.
(16)251: “other carbohydrates”, lignin is not a carbohydrate. Also, carbohydrate regions not shown, so no indication of residual carbohydrates provided
Reply: “other carbohydrates” has already been deleted.
(17)267-268: What does the difference in the spectra mean about the structure of the lignin?
Reply: A clear signal of C2,6−H2,6 bonds of the S unit located in δC/δH 103.93/6.69 ppm, appeared in Re-L, but not in R2. This distinction may be due to conditional differences between the two reactions. We think that this is due to the more acidic water solution of ZnCl2 and the weaker acidity of the system in DES, so this signal is retained in Re-L.
(18)283: No equivalent diagram to fig. 11 for the DES treatment? & 290: What is the benefit of the DES system? Industrially, how would you suggested recovering DES from this? & 351: any data on losses to the solvent system of low molecular weight material?
Reply: Thank you for your suggestion. In this work, we demonstrated the feasibility of selective degradation of hemicellulose by zinc chloride solution and then obtained cellulose and lignin respectively by the DES process. In this complete process, we propose the pretreatment of hemicellulose with a suitable concentration of zinc chloride solution followed by treatment of R1 using DES. [3]DES can be reused at least 5 times, in each process in addition to 30% of lactic acid and glycol to balance the loss. A detailed discussion and application of this DES fractionation process will be presented in our next work.
(19)295: Which LAP method (I assume LAP 15?) was used? With what column? Why was this not used for analysis of xylose / glucose / furfural (which can be analyzed in one go on a hydrogen counter-ion ion exclusion column with dilute sulfuric acid as eluent). & 405: how was the material utilized (no utilization was done in this paper)
Reply: Thank you for your suggestion. The approach we have taken is” Determination of Structural Carbohydrates and Lignin in Biomass Laboratory Analytical Procedure (LAP)”(2008.4.25). In this work, we focused more on the separation of xylose from glucose (we used High-performance liquid chromatography to determine the yield). For furfural, less than 0.2% of the total amount of furfural in the product was determined using High-performance liquid chromatography, which was not a concern in this work.
(20)315-317: lack of clarity on which hplc methods (I can guess, but I should not need to) & 334-335: what HPLC method? &382: I have never heard of (nor can I find) a “1M NH2 column”. Please give manufacturer and model / part number.&.385: HPLC manufacturer and model ideally should be include. Column manufacturer and model are important for replication.
Reply: Information about the HPLC method and column has been added to the original 3.11 & 3.12, details are as follows:
“3.11. Determination of xylose and glucose in the filtrate L1, L2 by means of HPLC
Monosaccharides in the filtrates were quantitatively analyzed by means of high-performance liquid chromatography (Shimadzu) equipped with a refractive index detector (RID-10A) and NH2 column (4.6 × 250 mm2, purchased from Agilent, 880952-708) at 40 °C. The mixed solution of acetonitrile and deionized water (70:30, v/v) was used as the mobile phase at a flow rate of 1.0 mL min−1.
3.12. Determination of furfural and 5-HMF in the C1 by means of HPLC
The dehydration products (5-HMF and furfural) of the monosaccharides were determined using HPLC(Shimadzu) with a C18 column (4.6 × 250 mm2, purchased from Agilent) at 40 °C and a UV−vis detector(SPD-20A) at 210nm. The mixed solution of methanol and ultrapure water (30:70, v/v) was used as the mobile phase at a flow rate of 0.7 mL min−1. Due to the very low content, only furfural and 5-HMF were qualitatively examined in this work.”
(21)317: why add activated carbon?
Reply: Thank you for your suggestion. The activated carbon is used to de-color the corncob in the filtrate. Before the adsorption, the filtrate is dark yellow, and then the filtrate is colorless. Activated carbon is used because it is cheap, easily available, environmentally friendly, has good absorption of pigments, and is easy to filter out.
(22)319: what kind of ammonia. Volume? conditions?
Reply: Thank you for your suggestion. Ammonia gas, analytical pure, purchased from Anhui Hefei Hengxing Industrial Gas Co., Ltd. , stored in cylinders, the flow rate is 1ml/s. This information has been added to the original text 3.1.
(23)322: xylose was distilled or dried? & 340: Glucose was distilled or dried?
Reply: We removed the water from the last filtrate by vacuum distillation. The xylose or glucose was obtained as a solid and kept in a flask without air.
(24)354: how was water solubility measured by XRD?
Reply: The solid Zn(OH)2 standard materials and S1, S2 was characterized by XRD to determine the existing forms of the recovered zinc ions and the main components of S1, S2.
(25)362-364: this is confusing and unclear. &401: Yield and selectivity appear to be getting used interchangeably, but with differing definitions. Define and use precisely please.
Reply: Thank you for your suggestion. Selectivity is the ratio of xylose or glucose to the total mass of monosaccharides, with the denominator being the sum of the mass of xylose and glucose. For example, in 2.2, Table2,95 °C, 10 min, the yield of xylose and glucose obtained by the first reaction was 30.4% and 3.6%, respectively, and the selectivity of xylose was 30.4%/(30.4% + 3.6%).
Referencees
- Su, H.; Wang, J.; Yan, L. Homogeneously Synchronous Degradation of Chitin into Carbon Dots and Organic Acids in Aqueous Solution. ACS Sustainable Chemistry & Engineering 2019, 7, 18476-18482, doi:10.1021/acssuschemeng.9b04436.
- Xia, C.; Zhu, S.; Feng, T.; Yang, M.; Yang, B. Evolution and Synthesis of Carbon Dots: From Carbon Dots to Carbonized Polymer Dots. Adv Sci (Weinh) 2019, 6, 1901316, doi:10.1002/advs.201901316.
- Yu, Y.; Cheng, W.; Li, Y.; Wang, T.; Xia, Q.; Liu, Y.; Yu, H. Tailored one-pot lignocellulose fractionation to maximize biorefinery toward versatile xylochemicals and nanomaterials. Green Chemistry 2022, 24, 3257-3268, doi:10.1039/d2gc00264g.
- <Determination of Structural Carbohydrates and Lignin in Biomass Laboratory Analytical Procedure (LAP).pdf>.
Round 2
Reviewer 1 Report
The authors addressed previous reviewer concerns. The revised manuscript is suitable for publication.
Author Response
Thank you for your suggestions and work, which we think contributed to this article. We use the"Track change" feature in Microsoft Glossary to edit the original text. Here are the answers to your questions. We mark the second supplementary response in red and the first in blue in our response to the reviewers' letter so that you can identify it.
(1)I see several superscript zero insted of a proper degree sign. This must be fixed. I also note that the authors reply (quite good answers however) to the reviewer's commentsm without changing anything in the MS. Sometimes this must be done.
Reply: Thank you for your suggestion. We have checked and corrected the symbols for degrees. We revised the original text at the suggestion of the reviewers and marked the second revision with a “Track Change” pattern in MS Word.
(2)The ammount of HCl/water (of no water at all) of affect the pH VERY MUCH. I have not, however, found any discussion about this. This is, of course, very imortant, I can, however, have missed that.
Reply: Thank you for your suggestion. We very much agree with you that the concentration of hydrochloric acid is very important for the amount of this reaction, the lower concentration will not be conducive to the degradation of biomass, and a high concentration is prone to over-reaction and carbonization. In this work, the dosage of hydrochloric acid concentration is mainly based on our team's previous work[1], so no more details. In this paper, the amount of hydrochloric acid used for each reaction has been clearly indicated, and we used a standard concentration of concentrated hydrochloric acid and deionized water configured as 2 mol/L hydrochloric acid for use, it is also our expectation that this experiment and phenomenon will be repeated by reviewers and readers.
(3)Also very THE AUTHORS must important, is the reviewer's comment about the moisture. I quote: " Moisture needs to be accounted for in the biomass and products. How biomass was stored to keep it dry or moisture constant is not stated. ". I don´t find any such comments at all. This must be fixed if publication is to be considered.”
Reply: Thank you for your suggestion. On line 299, We have added a description of how corncobs are stored “Corncob from Inner Mongolia, China, were placed in sealed plastic bags to dry naturally for about three months, and ground to powder and screened through 80 mesh before being used. The weight loss of corncob powder was less than 0.01 g/g in three months, and powder contains of 7.4% water still.”
Referee:2#
Comments and Suggestions for Authors
Interesting direction of research. As noted in my review document, there is considerable lack of clarity on several points. The methods used need to be clarified. Moisture needs to be accounted for in the biomass and products. How biomass was stored to keep it dry or moisture constant is not stated. I recommend clarifying what figures 2 and 3 are trying to say, as it is not clear. Having them next to each other makes it much less clear that figure 3 is specifically just for the DES process. Finally, I strongly recommend giving some thought to seeming to claim industrial value, without a clear analysis of process chemical recovery.
Reply: Thanks, we are appreciated to your positive comments. Thank you very much for your careful reading and advice. I think it is very helpful for the work. We have corrected the grammatical and expressive mistakes you mentioned. Here are the answers to the questions and their modifications. Thanks Again!
- 17 & 19: degree signs are not correct
Reply: Thank you for your suggestion. This error has been corrected on lines 17-19 of the original article.
- 20: wording of “yield of glucose are” is off
Reply: Thank you for your suggestion. On line 20, we have corrected “the yield of glucose are 29.4w% (selectivity=92%).”to “29.4w% (selectivity=92%) of glucose can be obtained.”
3)31: odd word choice of “forced”
Reply: Thank you for your suggestion. On line 31, we have corrected “forced” to “prompted”.
(4) 38: “loose irregular structure”, but no comment on the weaker nature of the xylose-xylose bond, or the accessibility of hemicelluloses due to their position in the biomass?
Reply: Thank you for your suggestion. On line 38, we have corrected “hemicellulose was more susceptible to degradation and separation due to loose and irregular structures.” to “Compared with cellulose, hemicellulose has fewer hydrogen bonds and exposes more active centers for degradation.” And we inserted relevant references.[2]
(5) 51: Corncobs are not a representative biomass (should be lignocellulosic biomass, not lignocellulose). They have a very hemicellulose content, much higher than almost any other biomass I am familiar with. It is a biomass of interest, but no representative.
Reply: Thank you for your suggestion. On line 51, we have corrected “Corncob is a representative lignocellulose,.” to “Corncob is a lignocellulosic biomass of interest,”.
(6)56: Wording of “such as alkaline extraction”& 56: Alkaline systems in pulp and paper industry show low rates of corrosion on common stainless steels,and carbon steel has been used in many older facilities. Also, alkaline treatments do not generate sugars from degradation usually, they form salts of sugar acids (pulping “peeling” reactions) while solubilizing polymer chunks.
Reply: Thank you for your suggestion. On line 54-55, we have corrected “to produce xylose and glucose using biomass,” to “for the treatment and degradation of biomass,”.
On line 56-57, we have corrected “Such as alkaline extraction can partially degrade carbohydrates and produce some phenolic derivatives but causes severe corrosion of the equipment and pollution of the environment.” to “Alkaline systems can partially degrade carbohydrates and produce some phenolic derivatives.”.
(7)65 (also 293-303): Moisture content of the corn cobs was not stated or controlled. Unclear if material is air dried, oven dried, or green, which can affect reactivity.
Reply: Thank you for your suggestion. On line 296, We have added a description of how corncobs are stored “Corncob from Inner Mongolia, China, were placed in sealed plastic bags to dry naturally for about three months, and ground to powder and screened through 80 mesh before being used. The weight loss of corncob powder was less than 0.01 g/g in three months.”.
(8) If this is focused on industrial processing, ammonia for recovery is not going to be cost effective (especially at the high zinc use rates), and the Cl ion is going to cause issues.
Reply: Thank you for your suggestion. In industry, the use of ammonia is common, and the low cost of industrial ammonia makes this process potential. With the further optimization of this method, we will focus on the recovery and reuse of Cl ions in the following work.
(9)74: wording of “selectively degradation” is off.
Reply: Thank you for your suggestion. On lines 72-74, we have corrected “It provides a simple and efficient method to selectively degradation of biomass to xylose, glucose, and lignin.” to “It provides a simple and efficient method of treating biomass to produce xylose, glucose, and lignin.”.
(10) 85 / table 1 (also 117, 119, tables 2&3, 155/table 4, 157 table 5,): Denominator for yields is unclear. Unclear that the units are in the first line / headings.
Reply: Thank you for your suggestion. On lines 117,119,155&157, we have indicated the denominator in the title of each of the tables you mentioned and colored it red.
(11) 86: the description of the solution is unclear. Is one condition different? Were volumes consistent? This does not seem consistent with section 3.2.
Reply: Thank you for your suggestion. Section 2.1 describes the effect of zinc chloride concentration on the yield of xylose and glucose. The yields of xylose and glucose were higher at higher concentrations, but only xylose was higher at lower concentrations. Therefore, we designed a two-step reaction scheme, the first step was treated with a lower concentration of solution to obtain a higher xylose yield, and the second step was treated with a higher concentration of solution to obtain a higher glucose yield. Sections 2.2 and 2.3 discuss the effects of temperature and time on the stepwise reaction, and section 3.2 and 3.3 describes the specific operation of the stepwise reaction.
(12) 86: up to this point it was unclear that this system is ZnCl2 + HCl. Suggesting this is not corrosive (comments in introduction about other things being corrosive), is misleading.
Reply: Thank you for your suggestion. On lines 47-48, we have removed comments about other methods that are corrosive. “Aqueous solutions of inorganic salts are less corrosive to the equipment as compared to acids, and can be easily recovered from the system,[24] while they are also more economically reasonable.”
(13) 90: caption incomplete.
Reply: Thank you for your suggestion. The caption has been completed.
(14) 91: refers to a “melted salt” but elsewhere treats it as a solution. This is confusing. & 91: this entire sentence appears to have not purpose.
Reply: Thank you for your suggestion. On lines 88-89, we have removed “Acidic ZnCl2 melted salt aqueous solution can efficiently degrade the polysaccharide to oligosaccharide and monosaccharides in a one-pot reaction.”.
(15) 96: The 50 wt.% is 50% of what? Is the denominator the solution? The biomass?
Reply: Thank you for your suggestion. The denominator is the total mass of the solution, which we have indicated in each wt% and marked in red.
(16) 101: wording of “structured more scattered” &101: wording of “interestingly, cellulose tightly packed structure”
Reply: Thank you for your suggestion. On lines 97-100, we have corrected “and the low concentration of treatment can only degrade the structured more scattered hemicellulose. Interestingly, cellulose tightly packed structure was also degraded when treated at high concentrations (75w%).” to “and the lower concentration of treatment can only degrade hemicellulose because there are fewer hydrogen bonds. Interestingly, due to the existence of a large number of hydrogen bonds, cellulose has a dense crystalline region, and was also degraded when treated at high concentrations (75w% of solution).”.
(17) 105: this is not a mechanism, it is a description of the methods used and naming conventions used.
Reply: Thank you for your suggestion. On line 104, we have corrected “mechanism” to “method”.
(18) 107: how was cellulose “better retained”, this is unclear, and may depend on a list of factors.
Reply: Thank you for your suggestion. On lines 107-108, we have corrected “while cellulose was better retained in R1” to “while cellulose was better retained in R1 which can be demonstrated by the XRD in Figure 6b”.
(19)110:Why are the filtrates being processed over activated carbon? What purpose does this have, and what is being absorbed?
Reply: Thank you for your suggestion. The activated carbon is used to adsorb the pigment from the corn cob in the filtrate. Before the adsorption, the filtrate is dark yellow, and then the filtrate is colorless
(20)110: how are you recovering xylose and glucose by “distillation”
Reply: Thank you for your suggestion. Water was removed by reduced-pressure distillation, and xylose and glucose were obtained as residual solids
(21) 110: spelling: glulose
Reply: Thank you for your suggestion. On line 110, we have corrected “glulose” to “glucose”.
(22)112: What this figure is referring to is unclear until much later in the paper. In this location it is extremely confusing.
Reply: Thank you for your suggestion. These symbols represent the products obtained in each process, and the article uses them to distinguish between them.
(23) 124: wording “efficiently hydrolysis”
Reply: Thank you for your suggestion. On line 124, we have removed “fully efficiently”.
(24) 125: wording “fully hydrolysis”
Reply: Thank you for your suggestion. On line 125, we have removed “fully”.
(25)126:“high concentration of acid” was additional HCl added?
Reply: Thank you for your suggestion. Because there is no added deionized water in the second step, the concentration of hydrochloric acid in the reaction is higher. The amount of used hydrochloric acid has indicated in the title of Table 2, 3, 4, and 5. We very much agree with you that the concentration of hydrochloric acid is very important for the amount of this reaction, the lower concentration will not be conducive to the degradation of biomass, and a high concentration is prone to over-reaction and carbonization. In this work, the dosage of hydrochloric acid concentration is mainly based on our team's previous work, so no more details. In this paper, the amount of hydrochloric acid used for each reaction has been clearly indicated, and we used a standard concentration of concentrated hydrochloric acid and deionized water configured as 2 mol/L hydrochloric acid for use, it is also our expectation that this experiment and phenomenon will be repeated by reviewers and readers.
(26)127: The concentrations of ZnCl2 used throughout are quite high. I would not consider any of them “low” & 128: “low concentration” of what?
Reply: Thank you for your suggestion. The concentration in the first step is lower than in the second step, so it is more appropriate to use the comparison level. "low" has been replaced with "lower" in the article
(27)129: (again at 130,139, 164, 170, 285, 288): weight % of what (biomass? Xylan? Theoretical xylose?)
Reply: Thank you for your suggestion. The denominator for all quality fractions is total corncob mass and has been restated in the relevant chart heading.
(28) 132: This sentence is unclear
Reply: Thank you for your suggestion. On line 132, we have removed “As the reaction temperature continues to rise, the reaction rate accelerates, while the primary reaction rate is greater than the side reaction rate, with the highest yield.”
(29) 133: What is meant by “continues to heat up”
Reply: Thank you for your suggestion. On line 133, we have corrected “then it continues to heat up,” to “then as the temperature continues to rise,”.
(30)135: “carbon dots”? what is meant by this, and is there data to support this?
Reply: Thank you for your suggestion. Carbon dots (CDs), as a new type of fluorescent carbon-based nanomaterial, have attracted considerable attention since first reported in 2004. The work of our team and others has demonstrated that the generation of carbon dots is related to the carbonization of biomass, and the relevant references have been inserted in the original text.[3,4]
(31) 138-139: “but also peaked”? This sentence is generally unclear. & 140-141: not clear how this ties in, may be due to lack of clarity in last sentence
Reply: Thank you for your suggestion. On line 138-139, we have corrected “but also peaked” to “the yield of glucose also peaked at 95°C (29.4w%).,”.
(32)143: How do you know no reaction occurred during the heat up with ZnCl2 and biomass?
Reply: Thank you for your suggestion. Because during the preheating of ZnCl2 and biomass, they are all solid, not in the same phase, and no liquid participates.
(33)143-148: what kind of water / oil bath and reaction vessel were used? 115 deg. C suggests a pressure flask and oil bath?
Reply: Thank you for your suggestion. A normal pressure oil bath heating pot and a 50ml glass round bottom flask were used, with no obvious boiling and vaporization at 115 degrees Celsius, this may because the boiling point of aqueous zinc chloride solution is higher than water which we thought it was feasible in terms of safety.
(34) 152: truth? Not clear what this means.
Reply: Thank you for your suggestion. we use preheating to ensure the accuracy of the reaction temperature, and in the experiment with the Mercury-in-glass thermometer multiple measurements were to prove its authenticity.
(35) 159: were listed? Wording
Reply: Thank you for your suggestion. On line 158, we have corrected “listed” to “shown on,”.
(36) 172: differing scales in SEM images make comparison difficult.
Reply: Thank you for your suggestion. In Figure5, b, c, and e are the comparison of R 1 and R 2 at 50 μ m on a scale, and d and f are the comparison of R 1 and R 2 at 30 μ m on a scale, respectively.
(37) 182: how was the biomass “utilized”?
Reply: Thank you for your suggestion. On line 132, we have removed “and utilizing.”
(38)186: MCC? This is never explained / sourced? Is this microcrystalline cellulose, and if so, why does it have a lignin peak in the FTIR?
Reply: Thank you for your suggestion. MCC is a microcrystalline cellulose purchased from Shanghai Aladdin reagent company, the source of which has been completed in this article on lines 305-307. You mentioned that it has a lignin absorption peak in FTIR. This may be due to the instrument operating conditions or the lack of thorough purification by the reagent company, but we do not think this affects the identification of its main features.
(39) 187: first?
Reply: Thank you for your suggestion. On line 158, we have corrected “First” to “The,”.
(40)211 (and 337): was aqueous or gaseous ammonia used? If aqueous, at what concentration
Reply: Thank you for your suggestion. The concentration of aqueous ammonia used is 25-28 w%, purchased from Sinopharm Chemical Reagents Co., LTD. (Shanghai, China), the source of which has been completed in this article on lines 302-303.
(41)223: “placing the filtrate” is unclear
Reply: Thank you for your suggestion. The filtrate was left standing in a 50 ml glass beaker for more than 24 h, and this information has been added to the original text on lines 221-222.
(42)223-225: I am unclear what is meant by this sentence
Reply: Thank you for your suggestion. A longer standing time allows more zinc ions to bind to ammonium ions and precipitate as a solid. Similarly, the concentration of zinc ions in the first filtrate is still high due to the precipitation-dissolution equilibrium. We add ammonia water or ammonia gas to each of the filtrates and repeat this many times to recover more zinc ions.
(43)226-228: These values of purity are inconsistent with the earlier yield numbers (Table 4&5)
Reply: Thank you for your suggestion. These purity values refer to the purity in the final obtained xylose and glucose crude product (obtained by subtracting the zinc ion mass in the last filtrate from the total mass of xylose or glucose), whereas Table 4 & 5, the yield refers to the value of xylose and glucose obtained with the quality of corncob as the denominator.
(44)234: No information is given on the method and equipment used for this analysis. NMR maker and MHz rating, probe, solvent system, preparation should be included or referenced.
Reply: Thank you for your suggestion. “For the 1H−13C HSQC NMR spectroscopy using a Bruker AVIII 400 MHz spectrometer equipped with a DCH cryoprobe, around 50 mg of Re-L or R2 was dissolved in 1.2 mL of DMSO-d6. The spectral widths were 12 ppm (from 11 to 1 ppm) and 220 ppm (from 200 to 20 ppm) for the 1H and 13C dimensions, respectively. The peak of the solvent (DMSO-d6) was used as a reference point for the internal chemical shift. Data of spectra were processed using standard MestReNova-NMR software.” This information has been added to the original text on line 232-237.
(45) 235: “peaks nearby” what?
Reply: Thank you for your suggestion. On line 158, we have corrected “signal peaks nearby” to “signals are displayed,”.
(46)251: “other carbohydrates”, lignin is not a carbohydrate. Also, carbohydrate regions not shown, so no indication of residual carbohydrates provided
Reply: Thank you for your suggestion. “other carbohydrates” has already been deleted on line 255.
(47)267-268: What does the difference in the spectra mean about the structure of the lignin?
Reply: Thank you for your suggestion. A clear signal of C2,6−H2,6 bonds of the S unit located in δC/δH 103.93/6.69 ppm, appeared in Re-L, but not in R2. This distinction may be due to conditional differences between the two reactions. We think that this is due to the more acidic water solution of ZnCl2 and the weaker acidity of the system in DES, so this signal is retained in Re-L.
(48)283: No equivalent diagram to fig. 11 for the DES treatment? & 290: What is the benefit of the DES system? Industrially, how would you suggested recovering DES from this? & 351: any data on losses to the solvent system of low molecular weight material?
Reply: Thank you for your suggestion. In this work, we demonstrated the feasibility of selective degradation of hemicellulose by zinc chloride solution and then obtained cellulose and lignin respectively by the DES process. In this complete process, we propose the pretreatment of hemicellulose with a suitable concentration of zinc chloride solution followed by treatment of R1 using DES. [5]DES can be reused at least 5 times, in each process in addition to 30% of lactic acid and glycol to balance the loss. A detailed discussion and application of this DES fractionation process will be presented in our next work.
(49)295: Which LAP method (I assume LAP 15?) was used? With what column? Why was this not used for analysis of xylose / glucose / furfural (which can be analyzed in one go on a hydrogen counter-ion ion exclusion column with dilute sulfuric acid as eluent). & 405: how was the material utilized (no utilization was done in this paper)
Reply: Thank you for your suggestion. The approach we have taken is” Determination of Structural Carbohydrates and Lignin in Biomass Laboratory Analytical Procedure (LAP)” (2008.4.25). In this work, we focused more on the separation of xylose from glucose (we used High-performance liquid chromatography to determine the yield). For furfural, less than 0.2% of the total amount of furfural in the product was determined using High-performance liquid chromatography, which was not a concern in this work.
(50)315-317: lack of clarity on which hplc methods (I can guess, but I should not need to) & 334-335: what HPLC method? &382: I have never heard of (nor can I find) a “1M NH2 column”. Please give manufacturer and model / part number.&.385: HPLC manufacturer and model ideally should be include. Column manufacturer and model are important for replication.
Reply: Thank you for your suggestion. Information about the HPLC method and column has been added to the original 3.11 & 3.12, details are as follows:
“3.11. Determination of xylose and glucose in the filtrate L1, L2 by means of HPLC
Monosaccharides in the filtrates were quantitatively analyzed by means of high-performance liquid chromatography (Shimadzu) equipped with a refractive index detector (RID-10A) and NH2 column (4.6 × 250 mm2, purchased from Agilent, 880952-708) at 40 °C. The mixed solution of acetonitrile and deionized water (70:30, v/v) was used as the mobile phase at a flow rate of 1.0 mL min−1.
3.12. Determination of furfural and 5-HMF in the C1 by means of HPLC
The dehydration products (5-HMF and furfural) of the monosaccharides were determined using HPLC(Shimadzu) with a C18 column (4.6 × 250 mm2, purchased from Agilent) at 40 °C and a UV−vis detector(SPD-20A) at 210nm. The mixed solution of methanol and ultrapure water (30:70, v/v) was used as the mobile phase at a flow rate of 0.7 mL min−1. Due to the very low content, only furfural and 5-HMF were qualitatively examined in this work.”
(51)317: why add activated carbon?
Reply: Thank you for your suggestion. The activated carbon is used to adsorb the pigment from the corn cob in the filtrate. Before the adsorption, the filtrate is dark yellow, and then the filtrate is colorless. Activated carbon is used because it is cheap, easily available, environmentally friendly, has a good absorption of pigments and is easy to filter out.
(52)319: what kind of ammonia. Volume? conditions?
Reply: Thank you for your suggestion. Ammonia gas, analytical pure, purchased from Anhui Hefei Hengxing Industrial Gas Co., Ltd., stored in cylinders, the flow rate is 1ml/s. This information has been added to the original text 3.1.
(53)322: xylose was distilled or dried? & 340: Glucose was distilled or dried?
Reply: Thank you for your suggestion. We removed the water from the last filtrate by vacuum distillation. The xylose or glucose was obtained as a solid and kept in a flask without air.
(54)354: how was water solubility measured by XRD?
Reply: Thank you for your suggestion. The solid Zn(OH)2 standard materials and S1, S2 were characterized by XRD to determine the existing forms of the recovered zinc ions and the main components of S1, S2.
(55)362-364: this is confusing and unclear. &401: Yield and selectivity appear to be getting used interchangeably, but with differing definitions. Define and use precisely please.
Reply: Thank you for your suggestion. Selectivity is the ratio of xylose or glucose to the total mass of monosaccharides, with the denominator being the sum of the mass of xylose and glucose. For example, in 2.2, Table2,95 °C, 10 min, the yield of xylose and glucose obtained by the first reaction was 30.4% and 3.6%, respectively, and the selectivity of xylose was 30.4%/(30.4% + 3.6%).
(56) 400: wording: “more easy to furthermore”
Reply: Thank you for your suggestion. On line 410, we have corrected “more easy to furthermore” to “easier,”
(57) 403: wording: “high pure”
Reply: Thank you for your suggestion. On line 413, we have corrected “high pure” to “higher purity,
(58) 405: how was the material utilized (no utilization was done in this paper)
Reply: Thank you for your suggestion. Conversion of biomass to xylose and glucose is an important utilization pathway, and we believe that the separation of cellulose and lignin from biomass is also the basis of utilization.
(59) 405: In my experience, this should be “lignocellulosic material” or “lignocellulosic biomass”
Reply: Thank you for your suggestion. On line 415, we have corrected “lignocellulose” to “lignocellulosic biomass”.
- Bi, Z.; Lai, B.; Zhao, Y.; Yan, L. Fast Disassembly of Lignocellulosic Biomass to Lignin and Sugars by Molten Salt Hydrate at Low Temperature for Overall Biorefinery. ACS Omega 2018, 3, 2984-2993, doi:10.1021/acsomega.8b00057.
- Jiang, Z.; Budarin, V.L.; Fan, J.; Remón, J.; Li, T.; Hu, C.; Clark, J.H. Sodium Chloride-Assisted Depolymerization of Xylo-oligomers to Xylose. ACS Sustainable Chemistry & Engineering 2018, 6, 4098-4104, doi:10.1021/acssuschemeng.7b04463.
- Su, H.; Wang, J.; Yan, L. Homogeneously Synchronous Degradation of Chitin into Carbon Dots and Organic Acids in Aqueous Solution. ACS Sustainable Chemistry & Engineering 2019, 7, 18476-18482, doi:10.1021/acssuschemeng.9b04436.
- Xia, C.; Zhu, S.; Feng, T.; Yang, M.; Yang, B. Evolution and Synthesis of Carbon Dots: From Carbon Dots to Carbonized Polymer Dots. Adv Sci (Weinh) 2019, 6, 1901316, doi:10.1002/advs.201901316.
- Yu, Y.; Cheng, W.; Li, Y.; Wang, T.; Xia, Q.; Liu, Y.; Yu, H. Tailored one-pot lignocellulose fractionation to maximize biorefinery toward versatile xylochemicals and nanomaterials. Green Chemistry 2022, 24, 3257-3268, doi:10.1039/d2gc00264g.
